# Factors and Inequality of Underweight and Overweight among Women of Reproductive Age in Myanmar: Evidence from the Demographic Health Survey 2015–2016

**Rajat Das Gupta [1], Mohammad Rifat Haider [2,\*] 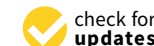 and Subhasish Das [3]**

[1]  Department of Epidemiology and Biostatistics, Arnold School of Public Health, University of South Carolina, Columbia, SC 29208, USA; rajatdas@email.sc.edu

[2]  Department of Social and Public Health, College of Health Sciences and Professions, Ohio University, Athens, OH 45701, USA

[3]  Nutrition and Clinical Services Division, International Centre for Diarrhoeal Disease Research, Bangladesh (icddr,b), 68 Shaheed Tajudddin Ahmed Sarani, Mohakhali, Dhaka 1212, Bangladesh; subhasish.das@icddrb.org

\*  Correspondence: haider@ohio.edu; Tel.: +1-740-566-8015

**Abstract:** (1) Background: This study aims to identify the factors of underweight and overweight/obesity among reproductive age (15–49 years) women in Myanmar, and assess the level of inequity in the double burden of malnutrition. (2) Methods: The study used Myanmar Demographic and Health Survey 2015–2016 data. Multinomial logistic regression models were fitted to identify the factors affecting underweight and overweight/obesity; and concentration indices (CI) were estimated to assess socioeconomic inequalities. (3) Results: A total of 12,643 reproductive age women were included in the analysis. Higher risk of underweight was found in women aged 20–29 years, aged 30–39 years, and 40–49 years compared to women aged 15–19 years; women who were unemployed or had manual occupation relative to those in non-manual employment. Women aged 40–49 years (compared to those who were 15–29 years); had primary education, and secondary education (compared to those who had no education); being married, and widowed/divorced/separated (compared to being never married); belonging to the poor quintile, middle quintile, richer, richest quintile (compared to the poorest quintile); having residence in urban areas (compared to rural areas) and in Kachin, Taninthayi, Yangon province (than those who lived in Naypyitaw province) had a higher risk of being overweight/obese. Socioeconomic inequalities were detected, with overweight/obesity strongly concentrated (CI: 0.19) amongst the higher quintiles and underweight concentrated (CI: −0.060) amongst the poorest. (4) Conclusions: Equity oriented nutrition interventions with a focus on improving the socioeconomic status of poor households may benefit undernourished women, while richer households should be focused to curb the overweight/obesity problem.

**Keywords:** underweight; overweight; determinants; Myanmar

## 1. Introduction

Over the last four decades, underweight as well as overweight and obesity have become a leading public health problem globally [1,2] The world has witnessed approximately five-fold increased prevalence of overweight and obesity among adult women and an eight-fold increased prevalence among adult men between 1975 and 2016 [3]. Within this time period, the highest rate was observed for the World Health Organization (WHO) South-East Asian region [3,4]. This is attributable to rapid

urbanization, lifestyles changes, and economic growth [4,5]. Additionally, the burden of overweight and obesity is higher among womenthan men in South Asia [6–8]. In the WHO South-East Asian region, one in four women are suffering from undernutrition and underweight, which is the highest in the world [9]. This co-existence of both undernutrition and over nutrition in the same setting is called a double burden of malnutrition [10].

Overweight and obesity is attributable to 2.8–4.0 million deaths globally. It is associated with cardiovascular diseases (CVD), hypertension, type 2 diabetes mellitus (T2DM), cancers, and other types of non-communicable diseases (NCDs) [11] and is responsible for 3.8% of disability-adjusted life years (DALYs) worldwide [12]. Overweight and obesity can adversely affect the female reproductive system. Increased risk of menstrual dysfunction and suppression of ovulation has been observed in overweight and obese women, which, in turn, make them susceptible to subfertility, infertility, and adverse pregnancy outcomes, including miscarriage and increased risk of cesarean section [13–18]. Maternal overweight and obesity leads to adverse effects on neonatal, infant, and child health [19]. Maternal underweight is also a risk factor of low birth weight and still birth in the offspring [20–22]. Additionally, maternal underweight leads to undernutrition in the offspring, which, in turn, results in NCDs like CVD, hypertension, and T2DM in adult life. This is a vicious cycle that continues through generations if left unchecked [23].

Myanmar, a South-East Asian nation, is facing a high burden of undernutrion and overnutrition. The Myanmar Demographic and Health Survey 2015–2016, that was conducted utilizing a nationally representative sample [24], found that one in seven women of reproductive age group was underweight (14.1%) [25]. The prevalence of overweight and obesity in the same group was around 40% [26]. Inequalities in the prevalence of overweight and obesity, as well as underweight, were observed in different South Asian Nations [27–29]. However, the pattern of inequality was not described in the context of Myanmar. We believe that understanding the pattern of inequality of the nutritional status of women of reproductive age group in Myanmar has the potential to guide policymakers and the program managers to design their interventions accordingly. Hence, this study aims to identify the inequality in the nutritional status of women of reproductive age (15–49 years) in Myanmar.

## 2. Materials and Methods

### 2.1. Study Design

This study used the Myanmar Demographic and Health Survey (MDHS) 2015–2016 data—the first edition of the survey conducted in Myanmar [24]. The cross-sectional survey was conducted through a joint collaboration between the Ministry of Health and Sports of Myanmar and ICF International. A nationally representative sample for this study was selected through a two-stage cluster sampling technique. In the first stage, 442 clusters (urban: 123 and rural: 319) were selected randomly from a sampling frame of 4000 clusters. From each of the 442 clusters, 30 households were selected for the second stage, and a total of 13,260 households were selected as the final sample. All women of reproductive age (15–49 years) residing permanently, or staying in the selected households the previous night of data collection, were interviewed under this survey. Around 96% of eligible women agreed to participate in the survey. Around 98% of the respondents agreed for anthropometric measurement. However, pregnant women and women who had given birth within the preceding 2 months of the survey were excluded. The final weighted sample size of this study was 12,643. Detail of the survey procedure can be found elsewhere [24].

### 2.2. Survey Tools and Data Collection

For capturing the local context, a standard women's questionnaire used by the DHS program was adopted, modified, and pretested, in order to collect socio-demographic, as well as reproductive health and nutrition-related information (e.g., age, gender, household wealth index, and place of residence) through face-to-face interviews. Interviews and anthropometric measurements were performed by

trained field staff. Measuring boards specially made by Shorr Productions were used for height measurement and lightweight SECA scales with digital screens were used for measuring the weight of the respondents.

The main outcome variables of this study were double burden of malnutrition, i.e., underweight and overweight/obese. To define these two variables, an Asia specific body mass index (BMI) cut-off value was used [30]. Women with a BMI < 18.5 kg/m$^2$ were considered to be underweight, women with a BMI between 18.5 kg/m$^2$ and <23.0 kg/m$^2$ were considered to be normal, and women with a BMI ≥ 23.0 kg/m$^2$ were considered to be overweight/obese.

Explanatory variables for this study include age (15–19 years, 20–29 years, 30–39 years, 40–49 years), education (no education, primary, secondary, college and higher), employment status (unemployed, non-manual, manual), marital status (never married, married, widowed/divorced/separated), media exposure (yes, no), wealth index (poorest, poorer, middle, richer, richest), place of residence (rural, urban), province of residence (Naypyitaw, Kachin, Kayah, Kayin, Chin, Sagaing, Taninthayi, Bago, Magway, Mon, Rahine, Yangon, Shan, Ayeyarwa). Media exposure was defined as anyone reporting viewing television, listening to radio, or reading a newspaper regularly or once a week. The DHS reports wealth index for each household using principal component analysis based on the ownership of various household assets and specific amenities of life [24].

## 2.3. Data Analysis

Sociodemographic characteristics of the respondents were reported using the weighted descriptive statistics (percentage with a 95% confidence interval (CI)). Bivariate tests (Chi-square) were performed to determine whether the respondents in different categories of explanatory variables differed in terms of BMI status. The significant variables ($p < 0.05$) in bivariate analysis were included in the multivariable analysis. In order to find the association between the explanatory and outcome variables, multinomial logistic regression analysis, with the normal weight as the base outcome, was performed. Multinomial logistic regression was used because the outcome variable had more than two categories. The variables that showed $p < 0.05$ in multivariable analyses were considered to be statistically significant. Both crude and adjusted odds ratios (OR) were reported. All analyses were weighted using women's individual sample weights.

The concentration curve shows the health inequality by plotting the cumulative percentage of health variable against the cumulative percentage of population ranked from poorest to the richest. If the concentration curve coincides with the 45-degree line, prefect equality is achieved. If the curve lies below the 45-degree line, it implies that the inequality is pro-rich; while, if the curve is located above the line, it implies pro-poor inequality. The concentration index (CI) is derived from the concentration curve and it measures the magnitude of inequality. CI is calculated as twice the area between the concentration curve and the line of equality, and the index value, by definition, is bound between −1 and +1. Zero value of the index indicates perfect equality, but if the index value becomes negative, it shows the disproportionate concentration of health variable among the poor, and if it becomes positive it implies the disproportionate concentration of health variable among the rich [31]. All analyses were performed using Stata 16.0 [32].

## 2.4. Ethical Consideration

The Ethics Review Committee on Medical Research including Human Subjects in the Department of Medical Research, Ministry of Health and Sports and ICF Institutional Review Board reviewed and approved the protocol of the MDHS [24]. Before data collection, written informed consent was taken from the respondents.

## 3. Results

### 3.1. Sociodemographic Characteristics of the Respondents

The overall prevalence of underweight was 15% (95% CI: 14.1–15.8), while the prevalence of overweight was 39% (95% CI: 37.6–40.5), and the prevalence of normal weight was 46% (95% CI: 44.7–47.3).

Table 1 reports the sociodemographic characteristics of the respondents and the prevalence of underweight and overweight/obesity among the respondents. Among the respondents, 30.9% were 30–39 years old, 41.4% had primary education, 47.0% were involved in manual labor, 60.4% were married, 86.9% had media exposure, 22.0% belonged to richest wealth quintile, 71.2% resided in the rural areas, and 14.1% were from Yangon province.

Table 1 also illustrates the distribution of underweight, normal weight, and overweight/obesity by age, education, employment status, marital status, media exposure, wealth quintile, residence, and province. Underweight was most prevalent in the 15–19 age group (26.7%), among respondents with secondary education (17.4%,), who were a manual laborer (16.7%), never married (23.2%), had media exposure (15.1%), in the poorest wealth quintile (19.3%), lived in rural areas (16.0%), and residents of Bago province (21.7%).

**Table 1.** Sociodemographic characteristics of the reproductive age women under the study and prevalence of underweight, normal weight, and overweight/obesity by socio-demographic characteristics (weighted *N* = 12,643), MDHS 2015-16.

| Characteristics | Total | | BMI (%) | | |
|---|---|---|---|---|---|
| | Percentage (%) | 95% CI | Underweight (%) | Normal Weight (%) | Overweight/ Obesity (%) |
| Age (in Years) *** | | | | | |
| 15–19 | 14.1 | 13.4–14.8 | 26.7 | 60.1 | 13.2 |
| 20–29 | 28.9 | 28.1–29.9 | 18.5 | 51.9 | 29.6 |
| 30–39 | 30.9 | 30.0–31.9 | 9.9 | 42.2 | 47.9 |
| 40–49 | 26.1 | 25.1–27.1 | 10.7 | 36.5 | 52.8 |
| Education *** | | | | | |
| No education | 12.5 | 11.0–14.2 | 14.6 | 50.5 | 34.9 |
| Primary education | 41.4 | 39.6–43.2 | 13.3 | 46.2 | 40.5 |
| Secondary education | 36.0 | 34.3–37.8 | 17.4 | 45.1 | 37.4 |
| College and higher | 10.1 | 9.0–11.3 | 13.3 | 43.0 | 43.7 |
| Employment Status *** | | | | | |
| Unemployment | 27.3 | 25.7–29.0 | 16.1 | 44.4 | 39.5 |
| Non-Manual | 25.7 | 23.9–27.5 | 10.5 | 40.3 | 49.2 |
| Manual | 47.0 | 44.8–49.3 | 16.7 | 50.2 | 33.1 |
| Marital Status *** | | | | | |
| Never married | 33.1 | 32.0–34.3 | 23.2 | 53.5 | 23.3 |
| Married | 60.4 | 59.2–61.6 | 10.5 | 41.8 | 47.7 |
| Widowed/Divorced/Separated | 6.5 | 5.9–7.1 | 13.7 | 47.3 | 39.0 |
| Media Exposure *** | | | | | |
| No | 13.1 | 11.7–14.6 | 14.1 | 52.2 | 33.7 |
| Yes | 86.9 | 85.5–88.3 | 15.1 | 45.1 | 39.8 |
| Wealth index *** | | | | | |
| Poorest | 17.7 | 16.0–19.4 | 19.3 | 54.5 | 26.2 |
| Poor | 18.8 | 17.5–20.2 | 14.9 | 51.5 | 33.6 |
| Middle | 20.7 | 19.2–22.2 | 15.9 | 45.3 | 38.8 |
| Richer | 20.9 | 19.2–22.7 | 13.4 | 43.5 | 43.1 |
| Richest | 22.0 | 19.9–24.2 | 12.0 | 37.6 | 50.4 |
| Place of residence *** | | | | | |
| Rural | 71.2 | 69.8–72.6 | 16.0 | 49.1 | 34.9 |
| Urban | 28.8 | 27.4–30.2 | 12.3 | 38.5 | 49.2 |

**Table 1.** *Cont.*

| Characteristics | Total | | BMI (%) | | |
|---|---|---|---|---|---|
| | Percentage (%) | 95% CI | Underweight (%) | Normal Weight (%) | Overweight/ Obesity (%) |
| Province of Residence *** | | | | | |
| Naypyitaw | 2.3 | 2.0–2.7 | 16.2 | 46.8 | 37.0 |
| Kachin | 2.9 | 2.4–3.4 | 10.0 | 43.4 | 46.6 |
| Kayah | 0.5 | 0.4–0.6 | 8.9 | 54.5 | 36.6 |
| Kayin | 2.4 | 2.1–2.6 | 13.0 | 46.2 | 40.8 |
| Chin | 0.8 | 0.7–0.9 | 8.6 | 61.1 | 30.3 |
| Sagaing | 11.1 | 10.3–11.9 | 12.9 | 45.8 | 41.3 |
| Taninthayi | 2.2 | 2.0–2.5 | 15.5 | 42.0 | 42.5 |
| Bago | 9.8 | 9.1–10.6 | 21.7 | 42.8 | 35.5 |
| Magway | 8.5 | 7.8–9.3 | 18.0 | 49.7 | 32.3 |
| Mandalay | 12.1 | 11.1–13.1 | 16.9 | 46.6 | 36.5 |
| Mon | 3.6 | 3.3–3.9 | 14.0 | 45.4 | 40.7 |
| Rakhine | 5.9 | 5.3–6.6 | 18.9 | 55.8 | 25.3 |
| Yangon | 14.9 | 13.8–16.1 | 11.7 | 37.8 | 50.5 |
| Shan | 10.2 | 9.1–11.4 | 7.9 | 51.3 | 40.9 |
| Ayeyarwa | 12.8 | 11.8–13.8 | 17.3 | 46.6 | 36.1 |

MDHS: Myanmar Demographic and Health Survey, *** *p*-value < 0.001, derived from chi-square test.

Overweight/obesity was most prevalent in the 40–49 age group (52.8%), among women with higher education (43.7%), in non-manual occupation (49.2%), married (47.7%), who had media exposure (39.8%), were in richest wealth quintile (50.4%), lived in urban areas (49.2%), and in Yangon province (50.5%).

*3.2. Correlates of Underweight and Overweight/Obesity*

Table 2 shows odds ratios (OR) of both crude and adjusted multinomial logistic regression models for underweight and overweight/obesity in comparison to normal weight women. According to the crude regression models, women with increasing age (20–29 years, 30–39 years, 40–49 years) in comparison to 15–19 years; had secondary education relative to who had no education; women who were unemployed or in a manual occupation relative to those in non-manual occupation; had media exposure relative to those who did not; and those who resided in Bago province relative to those who were from Naypyitaw, were at significantly higher risk of being underweight. Those who were married and widowed/divorced/separated compared to those who were never married; those who belonged to poor wealth quintile compared to those who were from poorest quintile; those who lived in Kayah, Chin, and Shan province compared to residents of Yangon province, were at lower risk of being underweight.

On the other hand, crude regressions showed that women aged 40–49 years compared to 15–19 years; women who had primary, secondary, and college and higher level of education compared to women with no education; married and widowed/divorced/separated relative to never married; women from the poor, middle, richer, and richest wealth quintiles compared to women from the poorest quintile; who lived in urban areas relative to rural residents; who lived in Kachin, and Yangon relative to those who lived in Naypyitaw, were at significantly higher risk of being overweight/obese. Women aged 20–29 and 30–39 years relative to 15–19 years; women who were unemployed or had a manual occupation relative to those in non-manual employment; women who had media exposure relative to those who did not; and women living in Chin, and Rakhine relative to women living in Naypyitaw, were at significantly lower risk of being overweight/obese.

**Table 2.** Crude and adjusted odds ratios for correlates of underweight and overweight/obesity in comparison with normal weight women (*N* = 12,643), MDHS 2015-16.

| Characteristics | Crude | | | | Adjusted | | | |
|---|---|---|---|---|---|---|---|---|
| | Underweight | | Overweight/Obesity | | Underweight | | Overweight/Obesity | |
| | Odds Ratio | 95% CI | Odds Ratio | 95% CI | Odds Ratio | 95% CI | Odds Ratio | 95% CI |
| Age (in Years) | | | | | | | | |
| 15–19 | 1.00 | - | 1.00 | - | 1.00 | - | 1.00 | - |
| 20–29 | 1.89 *** | 1.57–2.27 | 0.19 *** | 0.16–0.24 | 1.36 * | 1.06–1.74 | 0.26 *** | 0.20–0.32 |
| 30–39 | 1.51 *** | 1.28–1.79 | 0.50 *** | 0.45–0.57 | 1.39 ** | 1.15–1.67 | 0.54 *** | 0.47–0.62 |
| 40–49 | 1.24 * | 1.04–1.48 | 1.28 *** | 1.13–1.44 | 1.27 ** | 1.07–1.52 | 1.23 ** | 1.08–1.40 |
| Education | | | | | | | | |
| No education | 1.00 | - | 1.00 | - | 1.00 | - | 1.00 | - |
| Primary education | 0.99 | 0.79–1.25 | 1.27 ** | 1.11–1.45 | 0.80 | 0.64–1.00 | 1.20 * | 1.03–1.40 |
| Secondary education | 1.34 * | 1.06–1.68 | 1.20 * | 1.03–1.39 | 0.97 | 0.77–1.22 | 1.25 * | 1.02–1.51 |
| College and higher | 1.06 | 0.80–1.42 | 1.47 *** | 1.20–1.80 | 0.90 | 0.67–1.21 | 0.91 | 0.70–1.18 |
| Employment Status | | | | | | | | |
| Unemployment | 1.39 *** | 1.17–1.65 | 0.73 *** | 0.63–0.85 | 1.42 *** | 1.18–1.71 | 0.82 * | 0.70–0.97 |
| Non-Manual | 1.00 | - | 1.00 | - | 1.00 | - | 1.00 | - |
| Manual | 1.27 ** | 1.08–1.49 | 0.54 *** | 0.48–0.61 | 1.32 ** | 1.11–1.57 | 0.69 *** | 0.60–0.80 |
| Marital Status | | | | | | | | |
| Never married | 1.00 | - | 1.00 | - | 1.00 | - | 1.00 | - |
| Married | 0.58 *** | 0.51–0.67 | 2.63 *** | 2.35–2.94 | 0.62 *** | 0.52–0.73 | 2.05 *** | 1.79–2.35 |
| Widowed/Divorced/Separated | 0.66 ** | 0.50–0.88 | 1.90 *** | 1.56–2.31 | 0.75 | 0.55–1.02 | 1.29 * | 1.03–1.60 |
| Media Exposure | | | | | | | | |
| No | 1.00 | - | 1.00 | - | 1.00 | - | 1.00 | - |
| Yes | 1.24 * | 1.00–1.53 | 0.65 *** | 1.18–1.59 | 1.16 | 0.95–1.42 | 1.00 | 0.85–1.18 |
| Wealth Index | | | | | | | | |
| Poorest | 1.00 | - | 1.00 | - | 1.00 | - | 1.00 | - |
| Poor | 0.81 * | 0.67–0.99 | 1.36 *** | 1.18–1.57 | 0.77 * | 0.63–0.94 | 1.32 ** | 1.12–1.55 |
| Middle | 0.99 | 0.80–1.23 | 1.78 *** | 1.52–2.10 | 0.87 | 0.70–1.08 | 1.76 *** | 1.47–2.11 |
| Richer | 0.87 | 0.71–1.06 | 2.07 *** | 1.75–2.45 | 0.76 * | 0.62–0.95 | 1.91 *** | 1.58–2.30 |
| Richest | 0.90 | 0.73–1.11 | 2.79 *** | 2.35–3.31 | 0.79 | 0.61–1.04 | 2.32 *** | 1.87–2.88 |
| Place of Residence | | | | | | | | |
| Rural | 1.00 | - | 1.00 | - | 1.00 | - | 1.00 | - |
| Urban | 0.98 | 0.85–1.13 | 1.80 *** | 1.57–2.06 | 1.02 | 0.85–1.23 | 1.31 ** | 1.12–1.52 |
| Province of Residence | | | | | | | | |
| Naypyitaw | 1.00 | - | 1.00 | - | 1.00 | - | 1.00 | - |
| Kachin | 0.67 | 0.38–1.16 | 1.36 * | 1.05–1.76 | 0.65 | 0.37–1.13 | 1.35 * | 1.03–1.77 |
| Kayah | 0.47 *** | 0.31–0.71 | 0.85 | 0.61–1.18 | 0.45 *** | 0.30–0.69 | 0.89 | 0.64–1.24 |
| Kayin | 0.82 | 0.56–1.19 | 1.12 | 0.84–1.49 | 0.80 | 0.54–1.18 | 1.13 | 0.86–1.50 |
| Chin | 0.41 *** | 0.27–0.61 | 0.63 ** | 0.45–0.87 | 0.38 *** | 0.25–0.57 | 0.70 * | 0.50–0.98 |
| Sagaing | 0.81 | 0.57–1.17 | 1.14 | 0.86–1.51 | 0.80 | 0.55–1.15 | 1.15 | 0.85–1.55 |
| Taninthayi | 1.06 | 0.70–1.61 | 1.28 | 0.98–1.67 | 1.02 | 0.67–1.54 | 1.39 * | 1.03–1.88 |
| Bago | 1.49 * | 1.01–2.12 | 1.05 | 0.83–1.32 | 1.45 | 1.00–2.11 | 1.11 | 0.85–1.43 |
| Magway | 1.05 | 0.72–1.52 | 0.82 | 0.62–1.08 | 1.04 | 0.71–1.53 | 0.88 | 0.66–1.19 |
| Mandalay | 1.05 | 0.73–1.51 | 0.99 | 0.76–1.28 | 1.03 | 0.71–1.49 | 0.99 | 0.74–1.31 |
| Mon | 0.89 | 0.60–1.31 | 1.13 | 0.86–1.50 | 0.84 | 0.57–1.26 | 1.11 | 0.80–1.54 |
| Rakhine | 0.98 | 0.67–1.42 | 0.57 *** | 0.42–0.78 | 0.91 | 0.62–1.33 | 0.80 | 0.57–1.11 |
| Yangon | 0.89 | 0.62–1.29 | 1.69 *** | 1.28–2.23 | 0.80 | 0.55–1.17 | 1.46 * | 1.08–1.96 |
| Shan | 0.44 ** | 0.28–0.71 | 1.01 | 0.76–1.34 | 0.43 ** | 0.26–0.70 | 1.16 | 0.87–1.55 |
| Ayeyarwa | 1.07 | 0.75–1.53 | 0.98 | 0.74–1.31 | 1.05 | 0.73–1.51 | 1.15 | 0.85–1.56 |

MDHS: Myanmar Demographic and Health Survey. * *p*-value < 0.05, ** *p*-value < 0.01, *** *p*-value < 0.001, derived from chi-square test.

Multivariable regression model shows that the odds of being underweight were higher among women aged 20–29 years (OR: 1.36, 95% CI: 1.06–1.74), aged 30–39 years (OR: 1.39, 95% CI: 1.15–1.67), 40–49 years (OR: 1.27, 95% CI: 1.07–1.52) compared to women aged 15–19 years; women who were unemployed (OR: 1.42; 95% CI: 1.19–1.71) or had a manual occupation (OR: 1.32, 95% CI: 1.11–1.57) relative to those in non-manual employment. While the risk of being underweight was lower among women who were married (OR: 0.62, 95% CI: 0.52–0.73) compared to those who were never married; women form the poor wealth quintile (OR: 0.77, 95% CI: 0.63–0.94) and the rich wealth quintile (OR: 0.76, 95% CI: 0.62–0.95) compared to those who were from poorest quintile; women living in Kayah (OR: 0.45, 95% CI: 0.30–0.69), Chin (OR: 0.38, 95% CI: 0.25–0.57), Shan (OR: 0.43, 95% CI: 0.26–0.70) compared to women living in Naypyitaw.

Whereas, in the adjusted model, the higher risk of being overweight/obese was observed among women aged 40–49 years (OR: 1.23, 95% CI: 1.08–1.40) than those who were 15–29 years; who had primary education (OR: 1.20, 95% CI: 1.03–1.40), and secondary education (OR: 1.25, 95% CI: 1.02–1.51) compared to those who had no education; women who were married (OR: 2.05, 95% CI: 1.79–2.35), and widowed/divorced/separated (OR: 1.29, 95% CI: 1.03–1.60) compared to women who were never married; women from the poor quintile (OR: 1.32, 95% CI: 1.12–1.55), middle quintile (OR: 1.76, 95%

CI: 1.47–2.11), richer (OR: 1.91, 95% CI: 1.58–2.30), and richest quintile (OR: 2.32, 95% CI: 1.87–2.88) compared to women from poorest quintile; women who lived in urban areas (OR: 1.31, 95% CI: 1.12–1.52) compared to women from rural areas; women living in Kachin (OR: 1.35, 95% CI: 1.03–1.77), Taninthayi (OR: 1.39, 95% CI: 1.03–1.88), Yangon (OR: 1.46, 95% CI: 1.08–1.96) compared to those who lived in Naypytiaw province. The risk of being overweight/obese was lower among women aged 20–29 years (OR: 0.26, 95% CI: 0.20–0.32), aged 30–39 years (OR: 0.54, 95% CI: 0.47–0.62) compared to women aged 15–19 years; who were unemployed (OR: 0.82, 95% CI: 0.70–0.97) and had a manual occupation (OR: 0.69, 95% CI: 0.60–0.80) compared to women who had a non-manual occupation; women living in Chin (OR: 0.70, 95% CI: 0.50–0.98) compared to women living in Naypytiaw province.

### 3.3. Socioeconomic Inequalities in Underweight and Overweight/Obesity

Socioeconomic inequalities in underweight are shown in Table 3. The overall CI indicated a disproportionate prevalence of underweight among the poor participants (CI: −0.06, standard error (SE) of CI: 0.01). The poorest (Q1)-to-richest (Q5) ratio was 1.61. The inequalities in underweight were higher among 40–49 years old (CI: −0.13, SE: 0.02), widowed/divorced/separated (CI: −0.13, SE: 0.03), those residing in Kayin (CI: −0.09, SE: 0.04), Chin (CI: −0.09, SE: 0.02), and Ayeyarwa province (CI: −0.10, SE: 0.02). Only among the 15–19 years old, was the prevalence of overweight significantly higher among the richest (Q5) compared to the poorest (Q1) (CI: 0.07, SE: 0.03).

**Table 3.** Socioeconomic inequalities in underweight in Myanmar, MDHS 2015-16.

| Variables | Poorest (Q1) (%) | Richest (Q5) (%) | Q1–Q5 | Q1:Q5 | Concentration Index © | Standard Error (SE) |
|---|---|---|---|---|---|---|
| Total | 19.3 | 12.0 | 7.3 | 1.61 | −0.06 *** | 0.01 |
| Age (in Years) | | | | | | |
| 15–19 | 21.8 | 31.0 | −9.2 | 0.70 | 0.07 * | 0.03 |
| 20–29 | 20.4 | 15.3 | 5.1 | 1.33 | −0.05 ** | 0.02 |
| 30–39 | 13.9 | 7.2 | 6.7 | 1.93 | −0.06 ** | 0.01 |
| 40–49 | 24.4 | 4.6 | 19.8 | 5.30 | −0.13 *** | 0.02 |
| Education | | | | | | |
| No education | 19.5 | 10.0 | 9.5 | 1.95 | −0.1 *** | 0.03 |
| Primary education | 18.2 | 9.9 | 8.3 | 1.84 | −0.08 *** | 0.01 |
| Secondary education | 23.2 | 13.9 | 9.3 | 1.67 | −0.07 *** | 0.02 |
| College and higher | 22.5 | 10.8 | 11.7 | 2.08 | −0.1 *** | 0.03 |
| Employment Status | | | | | | |
| Unemployment | 18.1 | 13.6 | 4.5 | 1.33 | −0.05 ** | 0.02 |
| Non-Manual | 14.6 | 10.2 | 4.4 | 1.43 | 0.02 | 0.01 |
| Manual | 20.7 | 13.5 | 7.2 | 1.53 | −0.06 *** | 0.01 |
| Marital Status | | | | | | |
| Never married | 27.1 | 20.0 | 7.1 | 1.36 | −0.05 ** | 0.02 |
| Married | 16.6 | 5.8 | 10.8 | 2.86 | −0.1 *** | 0.01 |
| Widowed/Divorced/Separated | 23.4 | 8.8 | 14.6 | 2.66 | −0.13 *** | 0.03 |
| Media Exposure | | | | | | |
| No | 18.1 | 8.5 | 9.6 | 2.13 | −0.08 *** | 0.02 |
| Yes | 19.9 | 12.2 | 7.7 | 1.63 | −0.06 *** | 0.01 |
| Place of Residence | | | | | | |
| Rural | 19.3 | 14.3 | 5 | 1.35 | −0.05 *** | 0.01 |
| Urban | 20.7 | 11.3 | 9.4 | 1.83 | −0.04 * | 0.02 |

**Table 3.** *Cont.*

| Variables | Poorest (Q1) | Richest (Q5) | Q1–Q5 | Q1:Q5 | Concentration Index © | Standard Error (SE) |
|---|---|---|---|---|---|---|
| | (%) | (%) | | | | |
| Province of Residence | | | | | | |
| Naypyitaw | 18.8 | 12.9 | 5.9 | 1.46 | −0.06 | 0.04 |
| Kachin | 13.5 | 6.3 | 7.2 | 2.14 | −0.04 | 0.03 |
| Kayah | 2.6 | 8.5 | −5.9 | 0.31 | 0.03 | 0.02 |
| Kayin | 19.3 | 8.9 | 10.4 | 2.17 | −0.09 * | 0.04 |
| Chin | 14.8 | 5.1 | 9.7 | 2.90 | −0.09 *** | 0.02 |
| Sagaing | 16.3 | 8.1 | 8.2 | 2.01 | −0.04 | 0.02 |
| Taninthayi | 15.2 | 19.4 | −4.2 | 0.78 | 0.02 | 0.04 |
| Bago | 27.4 | 12.4 | 15 | 2.21 | −0.08 * | 0.04 |
| Magway | 23.7 | 16.7 | 7 | 1.42 | −0.04 | 0.03 |
| Mandalay | 24 | 15.4 | 8.6 | 1.56 | −0.07 * | 0.03 |
| Mon | 13.4 | 9.1 | 4.3 | 1.47 | 0.04 | 0.03 |
| Rakhine | 20.6 | 16.1 | 4.5 | 1.28 | −0.04 | 0.04 |
| Yangon | 17.9 | 10.4 | 7.5 | 1.72 | −0.04 * | 0.02 |
| Shan | 7.7 | 9.9 | −2.2 | 0.78 | 0.02 | 0.03 |
| Ayeyarwa | 21 | 13.8 | 7.2 | 1.52 | −0.1 *** | 0.02 |

MDHS: Myanmar Demographic and Health Survey. * *p*-value < 0.05, ** *p*-value < 0.01, *** *p*-value < 0.001, derived from chi-square test.

Socioeconomic inequalities in overweight/obesity are shown in Table 4. The CI for overweight/obesity was 0.19 (SE: 0.01; *p* < 0.001), which indicates that the higher wealth quintiles bear the burden of overweight/obesity. The richest (Q5)-to-poorest (Q5) ratio was 1.92. The inequalities in overweight was higher among 40–49 years old (CI: 0.34, SE: 0.02), who received no formal education (CI: 0.26, SE: 0.03), married (CI: 0.27, SE: 0.02), widowed/divorced/separated (CI: 0.34, SE: 0.05), residing in Naypyitaw (CI: 0.28, SE: 0.03), Chin (CI: 0.33, SE: 0.03), and Rakhine province (CI: 0.25, SE: 0.03).

**Table 4.** Socioeconomic inequalities in overweight/obesity in Myanmar, MDHS 2015-16.

| Variables | Poorest (Q1) | Richest (Q5) | Q5–Q1 | Q5:Q1 | Concentration Ind©(C) | Standard Error (SE) |
|---|---|---|---|---|---|---|
| | (%) | (%) | | | | |
| Total | 26.2 | 50.4 | 24.2 | 1.92 | 0.19 *** | 0.01 |
| Age (in Years) | | | | | | |
| 15–19 | 10.1 | 18.0 | 7.9 | 1.78 | 0.07 * | 0.03 |
| 20–29 | 24.5 | 35.3 | 10.8 | 1.44 | 0.08 *** | 0.02 |
| 30–39 | 33.7 | 59.0 | 25.3 | 1.75 | 0.22 *** | 0.02 |
| 40–49 | 27.5 | 72.4 | 44.9 | 2.63 | 0.34 *** | 0.02 |
| Education | | | | | | |
| No education | 23 | 57.5 | 34.5 | 2.50 | 0.26 *** | 0.03 |
| Primary education | 28.7 | 61.1 | 32.4 | 2.13 | 0.24 *** | 0.02 |
| Secondary education | 23.1 | 47.6 | 24.5 | 2.06 | 0.17 *** | 0.02 |
| College and higher | 5.1 | 47.4 | 42.3 | 9.29 | 0.16 *** | 0.04 |
| Employment Status | | | | | | |
| Unemployment | 28.5 | 49.2 | 20.7 | 1.73 | 0.18 *** | 0.03 |
| Non-Manual | 31.3 | 52.4 | 21.1 | 1.67 | 0.09 *** | 0.03 |
| Manual | 24.3 | 47.0 | 22.7 | 1.93 | 0.15 *** | 0.02 |
| Marital Status | | | | | | |
| Never married | 10.3 | 31.9 | 21.6 | 3.10 | 0.16 *** | 0.02 |
| Married | 31.4 | 64.9 | 33.5 | 2.07 | 0.27 *** | 0.02 |
| Widowed/Divorced/Separated | 19.8 | 57.5 | 37.7 | 2.90 | 0.34 *** | 0.05 |
| Media Exposure | | | | | | |
| No | 23.4 | 49.7 | 26.3 | 2.12 | 0.23 *** | 0.03 |
| Yes | 27.4 | 50.4 | 23.0 | 1.84 | 0.18 *** | 0.02 |
| Place of Residence | | | | | | |
| Rural | 26 | 45.2 | 19.2 | 1.74 | 0.14 *** | 0.02 |
| Urban | 30.4 | 52.1 | 21.7 | 1.71 | 0.1 *** | 0.03 |

Table 4. *Cont.*

| Variables | Poorest (Q1) (%) | Richest (Q5) (%) | Q5–Q1 | Q5:Q1 | Concentration Ind©(C) | Standard Error (SE) |
|---|---|---|---|---|---|---|
| Province of Residence | | | | | | |
| Naypyitaw | 19.9 | 52.2 | 32.3 | 2.62 | 0.28 *** | 0.03 |
| Kachin | 33.3 | 55.9 | 22.6 | 1.68 | 0.13 * | 0.05 |
| Kayah | 29.3 | 46.0 | 16.7 | 1.57 | 0.14 * | 0.06 |
| Kayin | 26.5 | 51.4 | 24.9 | 1.94 | 0.23 *** | 0.04 |
| Chin | 11.9 | 59.2 | 47.3 | 4.97 | 0.33 *** | 0.03 |
| Sagaing | 27.3 | 59.0 | 31.7 | 2.16 | 0.17 *** | 0.04 |
| Taninthayi | 33.9 | 44.2 | 10.3 | 1.30 | 0.1 * | 0.04 |
| Bago | 24 | 45.5 | 21.5 | 1.90 | 0.16 *** | 0.04 |
| Magway | 22 | 37.0 | 15.0 | 1.68 | 0.1 * | 0.04 |
| Mandalay | 24.3 | 45.7 | 21.4 | 1.88 | 0.18 *** | 0.04 |
| Mon | 34.2 | 40.4 | 6.2 | 1.18 | 0.05 | 0.05 |
| Rakhine | 16.4 | 51.9 | 35.5 | 3.16 | 0.25 *** | 0.03 |
| Yangon | 40.4 | 51.5 | 11.1 | 1.27 | 0.1 * | 0.05 |
| Shan | 23.7 | 54.8 | 31.1 | 2.31 | 0.23 *** | 0.05 |
| Ayeyarwa | 31.4 | 40.0 | 8.6 | 1.27 | 0.13 *** | 0.03 |

MDHS: Myanmar Demographic and Health Survey. * *p*-value < 0.05, *** *p*-value < 0.001, derived from chi-square test.

## 4. Discussion

This study aimed to identify the correlates of double burden of malnutrition among the women of reproductive age group in Myanmar and assess its socio-economic inequalities. It was found that age of the women, employment status, marital status, wealth index, and province of residence was associated with both underweight and overweight/obesity. Education and place of residence were associated with overweight/obesity only. Furthermore, socio-economic inequalities were observed in the case of both underweight and overweight/obesity. Underweight was concentrated in the poorer socio-economic quintiles and overweight/obesity in the richer socio-economic quintiles. This is similar to the CI values found in Bangladesh and Nepal [26,31]. However, the CI value is small, and should be interpreted with caution.

This study found that the risk of underweight increased and the risk of overweight increased with the increasing age of the respondents. This finding is important for policymakers and public health nutrition managers for designing targeted interventions, in order to prevent and manage both underweight and overweight/obesity. Education was positively associated with overweight and obesity. Women with primary and secondary education had higher likelihood of being overweight/obese. This finding is consistent with findings from neighboring India [32], Nepal [26], and Bangladesh [33]. All of those studies utilized data collected from nationally representative sample. This may be due to the fact that better educated women are engaged in more sedentary jobs, and are less involved in physical activity [26].

Higher socioeconomic condition was found to be positively associated with overweight and obesity and negatively associated with underweight, which is consistent with the findings from neighboring South-Asian Countries like Bangladesh, India, and Nepal [32,34]. This is because the members from the higher socioeconomic households are involved in sedentary lifestyle and consume more energy dense food, which results in weight gain. The opposite scenario can be observed in developed nations, where the poor consume nutrient-poor but energy dense food more and, therefore, are at risk of becoming overweight/obese [35,36].

Those involved in manual work and were unemployed had a higher risk of being undernourished and a lower risk of being overweight/obese. Unemployment leads to a decreased energy intake. Manual workers are involved in physical activities that promote weight loss [37,38]. Marital status was also found to be associated with both underweight and overweight/obesity. Married and widowed/divorced women were more likely to be overweight/obese compared to their never married counterparts. This might be

due to gestational weight gain of the never married women [32]. There are also two plausible social explanations for the positive association between marriage and high BMI. The first one suggests that the married couple takes nutrition dense food regularly, which leads to increase in BMI. The second hypothesis which is known as 'marriage market hypothesis' suggests that the married couple, particularly females, do not pay attention to their BMI after marriage, and allow the BMI to rise [39]. This hypothesis also explains the lower risk of being underweight was among women who were married.

Urban residence also had increased risk of being overweight/obese. Urban areas are a more obesogenic environment. In the urban area, there is an increased availability of junk food. Moreover, the urban residents tend to be less physically active. The high energy intake, coupled with less energy expenditure, makes urban residents more prone to being overweight/obese [24]. Regional differences were also observed in the risk of underweight and overweight/obesity. The residents of Yangon had a higher probability of being overweight/obese. Yangon has predominantly urban area, which puts the residents at risk of being overweight/obese [24]. It is interesting to observe that residents of Chin had a lower risk of both undernutrition and overnutrition. Further exploration is needed in order to identify the cause.

The findings of the study indicate a multilevel and multifaceted intervention is needed, in order to prevent double burden of malnutrition in Myanmar. Women from the poorer wealth quintiles should be targeted for underweight prevention activities. On the other hand, richer wealth quintiles should be targeted for overweight and obesity prevention programs. Poverty reduction interventions, such as the production of fruits and vegetables, can be beneficial. This can be achieved by simple training in home gardening, microfinance programs, and small business [40,41]. Empowering women through participatory groups is a strategy that has already proven effective in improving maternal, neonatal, and child health (MNCH) [42,43], so it can be regarded as an effective strategy for improving nutritional knowledge and promoting physical activity [44,45]. This will facilitate prevention of both undernutrition and overnutrition. Additionally, higher inequality of underweight among the poorest participants aged 40–49 years, widowed/divorced/separated, and residents of Kayin, Chin, and Ayeyarwa province should be addressed. Among the 15–19 year olds, the direction of inequality was towards the richest quintiles. This should be explored. Similarly, health promotion programs should focus on the disproportionate burden of overweight/obesity among the richest quintiles.

This study has several notable strengths. First, this study utilized a nationally representative sample. As a result, the findings of the study can be generalizable to the target population of Myanmar. Second, due to utilization of validated tools and calibrated instruments by MDHS, the generated estimates are more robust than any other study done in the context of Myanmar. Finally, as we have used concentration index, the findings are more robust in predicating socio-economic inequality.

However, the limitations of the study warrant discussion. First, MDHS was a cross-sectional survey. As a result, we cannot establish temporal relationship between the explanatory variables and the outcome variable. Second, due to absence of data, several important variables, such as food security and dietary diversity, could not be included in the final model. Third, MDHS did not collect data on household income and expenditure. Rather, it used wealth index as a proxy indicator. Fourth, MDHS collected data on 15–49 years old women. As a result, the distribution of undernutrition and overnutrition could not be obtained for women beyond this age group.

## 5. Conclusions

The study identified inequality in the prevalence of both underweight and overweight and obesity among Myanmar's reproductive age group women. While the prevalence of underweight was concentrated to the poorer wealth quintiles, the distribution of overweight and obesity was skewed towards the richer wealth quintiles. The public health programs in Myanmar should incorporate an equality-based approach. Women from the poor economic subgroup, especially the elderly ones, should be targeted, in order to reduce the prevalence of undernutrition. On the other hand, women from the higher economic status should be targeted for overweight and obesity prevention

interventions. Additionally, women who are ever married, highly educated, living in an urban area, and residing in Yangon, Kachin, and Taninthayi should also be prioritized for the overweight and obesity prevention program.

**Author Contributions:** Conceptualization, R.D.G. and M.R.H.; methodology, R.D.G., M.R.H. and S.D.; software, M.R.H.; validation, R.D.G. and S.D.; formal analysis, M.R.H.; investigation, R.D.G.; resources, R.D.G., M.R.H. and S.D.; data curation, M.R.H.; writing—original draft preparation, R.D.G.; writing—review and editing, M.R.H. and S.D.; visualization, M.R.H.; supervision, M.R.H. and S.D.; project administration, R.D.G.; funding acquisition, not available. All authors have read and agreed to the published version of the manuscript.

**Funding:** This research received no external funding.

**Acknowledgments:** This study was conducted using the datasets of the Myanmar Demographic Health Survey (MDHS) 2015–2016. Hence, the authors of this study are thankful to DHS programme for offering the datasets.

**Conflicts of Interest:** The authors declare no conflict of interest.

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
