# Peer review of "Factors and Inequality of Underweight and Overweight among Women of Reproductive Age in Myanmar: Evidence from the Demographic Health Survey 2015–2016"

_epidemiologia, doi:10.3390/epidemiologia1010006_

Round 1

Reviewer 1 Report

Abstract

  1. In lines 16-17, perhaps the word “of” should come after the word “level”
  2. The presentation of the determinants in the abstract is quite difficult to follow for this reviewer. Perhaps the authors should consider for example simpling stating that women with higher education were more likely to be overweight/obesity and a higher household wealth index quintile compared to the poorest quintile was associated with overweight/obesity

Introduction

  1. A well-written introduction

Materials and methods

  1. Please check lines 96-97 for the repetitive phrase “women having a BMI between”
  2. In lines 117, the authors state that ” Both crude and adjusted relative risk rates (RRR) “ while the statistical method used was multinomial logistic regression. This is repeated through the manuscript but this reviewer thinks that the RRR should rather be odds ratios. The use of the term odds actually appear for the first time in 245
  3. In Table 1, this reviewer does not understand if the statistical significance test applies to all the outcomes presented or only one

Results

Clear presentation

Discission

  1. In lines 241-242, the authors write that “underweight increased and the risk of overweight decreased with increasing age of the respondents”. However, Table 2 shows that the odds of overweight was also higher (23%) among 40-49 yrs women and Table 1 also shows that this group also had the highest prevalence of overweight. The authors may consider re-writing this sentence or reproduce their estimates with age as a continuous variable to ascertain this
  2. In the discussion, I missed a possible explanation for why the risk of being underweight was lower among women who were married

Author Response

Abstract

  1. In lines 16-17, perhaps the word “of” should come after the word “level”

Response: Thank you for your comment. We have revised it.

  1. The presentation of the determinants in the abstract is quite difficult to follow for this reviewer. Perhaps the authors should consider for example simpling stating that women with higher education were more likely to be overweight/obesity and a higher household wealth index quintile compared to the poorest quintile was associated with overweight/obesity

Response: Thank you. We have revised this as following: “Women aged 40-49 years (than those who were 15-29 years); had primary education, and secondary education (than those who had no education); Being married, and widowed/divorced/separated (than being never married); belonging to poor quintile, middle quintile, richer, richest quintile (compared to poorest quintile); having residence in urban areas (compared to rural areas) and in Kachin, Taninthayi, Yangon province (than who lived in Naypytiaw province) had higher risk of being overweight/obese.”

Introduction

  1. A well-written introduction

Response: Thank you.

Materials and methods

  1. Please check lines 96-97 for the repetitive phrase “women having a BMI between”

Response: Thank you. We have deleted the repetitive phrase.

  1. In lines 117, the authors state that ” Both crude and adjusted relative risk rates (RRR) “ while the statistical method used was multinomial logistic regression. This is repeated through the manuscript but this reviewer thinks that the RRR should rather be odds ratios. The use of the term odds actually appear for the first time in 245

Response: Thank you for your comments. We agree that these Relative Risk Ratios (RRR) are actually Odds Ratios (OR) and we have made necessary changes throughout the manuscript.

  1. In Table 1, this reviewer does not understand if the statistical significance test applies to all the outcomes presented or only one

Response: Thank you for this question. The p-value was derived from chi-square test. It applies to all the variables.

Results

Clear presentation

Response: Thank you.

Discission

  1. In lines 241-242, the authors write that “underweight increased and the risk of overweight decreased with increasing age of the respondents”. However, Table 2 shows that the odds of overweight was also higher (23%) among 40-49 yrs women and Table 1 also shows that this group also had the highest prevalence of overweight. The authors may consider re-writing this sentence or reproduce their estimates with age as a continuous variable to ascertain this

Response: Thanks! We have revised the statement: “This study found that the risk of underweight increased and the risk of overweight increased with increasing age of the respondents.”

  1. In the discussion, I missed a possible explanation for why the risk of being underweight was lower among women who were married

Response: Thanks you for your comment. We added the following section: “There are also two plausible social explanation about the positive association between marriage and high BMI. The first one suggests that the married couple takes nutrition dense food regularly which leads to increase in BMI. The second hypothesis which is known as ‘marriage market hypothesis’ suggests that the married couple, particularly females do not pay attention to their BMI after marriage, and allow the BMI to rise [39]. This hypothesis also explains the lower risk of being underweight was among women who were married.”

Reviewer 2 Report

Dear Authors,

it is very nice study including a large number of participants. In the introduction you emphasize the importance of over and underweight (miscarriage and so on.) but have you compared your participants with such a risk?

what are the consequences of your participants considering the over or underweight?

if there is no comparison then the introduction should be changed, regarding DM, or cardiovascular disease or miscarriages 

Author Response

Dear Authors,

it is very nice study including a large number of participants. In the introduction you emphasize the importance of over and underweight (miscarriage and so on.) but have you compared your participants with such a risk?

what are the consequences of your participants considering the over or underweight?

if there is no comparison then the introduction should be changed, regarding DM, or cardiovascular disease or miscarriages 

Response: Thank you for the comment. In the introduction section we emphasize the importance of over and underweight by explaining the risk of underweight and overweight. This was written to provide the background as well as the rationale for the study. MDHS is a cross sectional study and did not collect the data on the consequences of underweight and overweight. Therefore, we did not have those variables in our model. And we humbly disagree with the reviewer that the introduction needs to be changed as the importance of over and underweight was described for justification.